# CDGraph: Dual Conditional Social Graph Synthesizing via Diffusion Model

## Abstract

The social graphs synthesized by the generative models are increasingly in demand due to data scarcity and concerns over user privacy. One of the key performance criteria for generating social networks is the fidelity to specified conditionals, such as users with certain membership and financial status. While recent diffusion models have shown remarkable performance in generating images, their effectiveness in synthesizing graphs has not yet been explored in the context of conditional social graphs. In this paper, we propose the first kind of conditional diffusion model for social networks, `CDGraph`, which trains and synthesizes graphs based on two specified conditions. We propose the co-evolution dependency in the denoising process of `CDGraph` to capture the mutual dependencies between the dual conditions and further incorporate social homophily and social contagion to preserve the connectivity between nodes while satisfying the specified conditions. Moreover, we introduce a novel classifier loss, which guides the training of the diffusion process through the mutual dependency of dual conditions. We evaluate `CDGraph` against four existing graph generative methods, i.e., SPECTRE, GSM, EDGE, and DiGress, on four datasets. Our results show that the generated graphs from `CDGraph` achieve much higher dual-conditional validity and lower discrepancy in various social network metrics than the baselines, thus demonstrating its proficiency in generating dual-conditional social graphs.

## 1 Introduction

Social networks offer a wide range of applications, such as viral marketing, friend recommendations, fake news detection, and more. However, achieving effective results often requires a substantial amount of personal data. Nevertheless, with the rise of privacy awareness, most individuals are reluctant to publicly disclose their personal information, including their profile and social interaction records, leading to a scarcity of data. The need of generating a social graph similar to the original one arises. It is critical for synthetic graphs to not only have similar structures as the original ones, e.g., centrality, but also satisfy exogenous conditions, such as specific user profiles.

Statistical sampling approaches (Shuai et al., 2018; Schweimer et al., 2022) have been used to produce graphs with certain social network properties, such as skewed degree distribution, a small diameter, and a large connected component, but they struggle to ensure the structure similarity to the original social graphs. Moreover, these methods cannot control the generation process to satisfy specified conditions (e.g., profiles of a social network user) (Bonifati et al., 2020). Recently, deep generative models are shown effective for synthesizing molecular graphs (Samanta et al., 2020; Chenthamarakshan et al., 2020), via extracting latent features from input graphs. The deep molecular graphs (Huang et al., 2022; Vignac et al., 2023) can well preserve the network structure and deal with a single exogenous condition only (e.g., chemical properties such as toxicity, acidity, etc.) but not more. They fail to capture the dependency between two specified conditions, and thus cannot generate graphs satisfying both conditions.

However, the generation of dual conditional social graphs is highly valuable for social applications, as these applications often require the consideration of users' diverse social contexts and connections, which typically satisfy more than one condition in real-world applications. Moreover, they should also retain the unique characteristics of social graphs, i.e., social homophily and social contagion, to ensure that the phenomena inherent in social applications can be captured. Social ho-

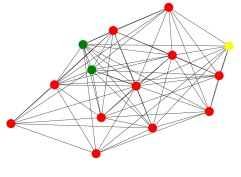

(a) Unconditional.

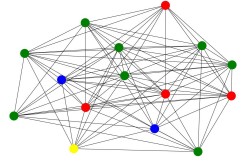

(b) Single-conditional.

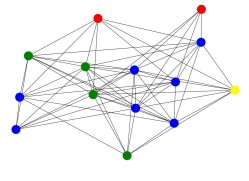

(c) Dual-conditional.

Figure 1: Illustrative examples of synthetic social graphs generated by unconditional, single-conditional, and dual-conditional approaches. The colors of the node represent the condition satisfaction: none (red), one (green and yellow), and both (blue).

mophily (Bisgin et al., 2012), which involves the tendency to form social connections with people who share similar profiles, plays a significant role as a primary factor in link formation (Lee et al., 2019). On the other hand, social contagion (Papachristou et al., 2023; Jiang et al., 2023), where behavior spontaneously spreads through a social graph, is largely influenced by network topology (Horsevad et al., 2022). Since homophily and contagion are generally intertwined [3], many studies often simultaneously consider both to facilitate social applications. For instance, Li et al. (Li et al., 2022) explore the effects of social homophily and contagion on users' behavior for social recommendation. Sankar et al. (Sankar et al., 2020) leverage social homophily and contagion to accurately predict the spread of information. For example, in social applications like social marketing, it is common to select individuals with strong influence within their social circles for word-of-mouth promotion. These social circles are parts of users' ego social graphs, preserving the essential characteristics of social homophily and social contagion to accurately capture social influence propagation. Specifically, for luxury golf club brands targeting golf enthusiasts and high-income individuals, analyzing users' influence within social circles that include friends sharing a passion for golf and possessing high incomes can help identify individuals well-suited for promoting the brands.

Generating social graphs satisfying dual conditions is no mean feat and faces several challenges. (i) *Intricate dependencies across conditions*: Those exogenous conditions are often mutually dependent, and modeling conditions independently may lead to sub-optimal synthetic graphs. For instance, there is a high correlation between being golf enthusiasts and income brackets. (ii) *Fulfilling graph structure similarity and exogenous conditions*: When synthesizing social graphs, one has to not only maintain the network structure but also adhere to the conditions. Compared to chemical graphs, where the structure is dictated primarily by physical and chemical constraints, social homophily and social contagion are prevalent phenomena in social graphs. That is, users' profiles subtly drive their social interactions and vice versa. Hence, it is crucial to follow the original structure to generate and link users based on their profiles while ensuring that dual conditions are met (i.e., avoiding generating an excessive number of users who do not meet the specified conditions). Figure 1 illustrates the generated graphs obtained by unconditional, single-conditional, and dual-conditional generative models. It can be observed that both unconditional and single-conditional methods fall short in generating graphs that satisfy dual conditions, all the while maintaining a network structure influenced by social homophily and social contagion.

In this paper, we propose a novel conditional diffusion model, *Dual Conditional Diffusion Graph (CDGraph)*, for synthesizing social graphs based on two exogenous conditions jointly. We first propose a novel notion of *co-evolution dependency* to capture the mutual dependencies between two exogenous conditions. In the conditional denoising process, we introduce the *co-evolution dependency* to bind the diffusion processes to the specified node conditions. Moreover, the co-evolution dependency is designed to account for *social homophily* and *social contagion*, which explore the dependencies between the nodes' associated conditions and connections. The Social homophily-based co-evolution ensures nodes with similar profiles are connected, while the social contagion-based co-evolution facilitates nodes connected with edges to share similar profiles. Equipped with the co-evolution dependency, CDGraph can preserve the connectivity between nodes that satisfy the specified conditions. Then, we design a novel loss function to train CDGraph with the guidance of the specified conditions, aiming to optimize the discrepancy in the co-evolution diffusion process. Furthermore, we introduce the notion of the *dual-condition classifier* that jointly steers the co-evolution diffusion process toward the estimated distributions of condition fulfillment. We evalu-

ate the performance of `CDGraph` on real social networks by measuring the dual-conditional validity and the discrepancy in various social network metrics. The contributions include:

- We justify the need to generate dual conditional social graphs exhibiting fundamental characteristics such as *social homophily* and *social contagion*. To address this requirement, we develop a novel dual conditional graph diffusion model, `CDGraph`.

- We propose *co-evolution dependency* not only to capture the interdependence between specified conditions but also to denoise edge and node embeddings based on social homophily and social contagion, respectively. Meanwhile, we introduce a *dual-condition classifier* to guide the denoising process, ensuring that the discrepancy in the diffusion process and condition fulfillment can be jointly optimized.

- Experiments conducted on four real-world social graphs demonstrate that CDGraph outperforms graph generation approaches based on traditional, diffusion-based, and deep generative models in various social metrics.

## 2 RELATED WORK

**Graph Generation.** The current research on graph generation techniques can be divided into statistic-based and deep generative model based ones. The existing statistic-based graph generation is mainly based on structural information such as network statistics (Schweimer et al., 2022), correlation (Erling et al., 2015), community structure (Luo et al., 2020), and node degree (Wang et al., 2021), etc. There are several social graph generators for various purposes, e.g., frequent patterns (Shuai et al., 2013) and similarity across social network providers (Shuai et al., 2018). For deep generative model-based ones, they are based on auto-regression (Liao et al., 2019; Shi et al., 2020), variational autoencoder (Guo et al., 2021; Samanta et al., 2020), and GAN (Martinkus et al., 2022), etc. Especially, SPECTRE (Martinkus et al., 2022) is a GAN-based conditional generative model conditioning on graph Laplacian eigenvectors. However, both statistical and deep generative methods do not consider dual conditions, since they mainly simply focus on structural conditions or dependencies between consecutive steps, instead of the connectivity between users satisfying specified conditions (i.e., linking them according to the original structure), while avoiding to generate excessive irrelevant users for meaningful downstream analysis such as estimating social influence of a user to a specified population.

**Diffusion Models.** The current research direction on diffusion models mainly focuses on the application to multimedia, such as computer vision, text-image processing, and audio processing (Saharia et al., 2022; Gu et al., 2022; Hoogeboom et al., 2021; Savinov et al., 2022). Denoising Diffusion Probabilistic Model (DDPM) (Ho et al., 2020; Dhariwal & Nichol, 2021) has performed significantly better than generative adversarial networks in image synthesis. Recently, diffusion models have been also applied to generate graph data (e.g., molecular graph generation) (Niu et al., 2020; Huang et al., 2022; Vignac et al., 2023; Chen et al., 2023) thanks to the flexible modeling architecture and tractable probabilistic distribution compared with the aforementioned deep generative model architectures. In particular, DiGress (Vignac et al., 2023) synthesizes molecular graphs with the discrete denoising probabilistic model building on a discrete space. It exploits regression guidance to lead the denoising process to generate graphs that meet the condition property. EDGE (Chen et al., 2023) is a discrete diffusion model exploiting graph sparsity to generate graphs conditioning on the change of node degree. However, the above existing studies only deal with a single condition on nodes or edges in graphs, failing to capture the dependencies of conditions on graphs.

## 3 CDGRAPH

In this section, we begin by introducing the dual conditional graph generation problem. Subsequently, we revisit the concepts of the discrete diffusion model with a single condition of DiGress (Vignac et al., 2023) as a preliminary to our approach. Finally, we propose `CDGraph`, featuring on the novel loss, the co-evolution dependency incorporating social contagion and social homophily, and the dual-condition classifier guidance.

We first provide the technical intuition behind `CDGraph`. To guide the diffusion process with dual conditions, an intuitive approach is to extend the conditional DiGress (Vignac et al., 2023) by adding

the guidance of the second condition. However, the dependencies across conditions are not explicitly captured and fall short in guiding the graph generation. Hence, we introduce the aforementioned dependencies to jointly optimize the structural similarity and condition satisfaction.

## 3.1 FORMAL PROBLEM DEFINITION

Here, we formulate the problem of *Dual Conditional Graph*. To perform the graph generation with the guidance of two conditions, we first define the *Condition Indication Graph* as follows. Figures 1(b) and 1(c) illustrate the two examples of conditional indication graphs.

**Definition 1** (Condition Indication Graph)*. Let $C$ denote the condition set. We define the condition indication graph of $C$ by $G_C = (\{\mathbf{X}_c\}_{c \in C}, \mathbf{E})$, where $\mathbf{X}_c \in \mathbb{R}^{N \times 2}$ indicate nodes' satisfaction of condition $c$, and $\mathbf{E} \in \mathbb{R}^{N \times N \times 2}$ indicates the existence of edges between nodes $v_n$ and $v_m$ with $\mathbf{e}_{n,m} = [0, 1]$ and $\mathbf{e}_{n,m} = [1, 0]$ otherwise. Specifically, $\mathbf{x}_{n,c} \in \{0, 1\}^2$ in $\mathbf{X}_c$ is a one-hot encoding vector representing whether a node $v_n$ in $G_C$ satisfies condition $c$; $e_{n,m} = (v_n, v_m)$ in $\mathbf{E}$ is a one-hot encoding vector representing whether an edge between $v_n$ and $v_m$ in $G_C$ satisfies condition c.*

**Definition 2** (Dual Conditional Graph Generation)*. Given the condition set $C = \{c_1, c_2\}$ and the condition indication graph $G_C = (\{\mathbf{X}_{c_1}, \mathbf{X}_{c_2}\}, \mathbf{E})$, the problem is to generate social graphs, such that i) the structural information of the generated graphs is similar to $G_C$, and ii) the majority of nodes in the generated graphs meet the conditions $c_1$ and $c_2$.*

## 3.2 PRELIMINARY: DISCRETE DIFFUSION MODELS FOR GRAPH GENERATION

We revisit DiGress (Vignac et al., 2023), which is a discrete diffusion model with a single condition, i.e., $C = \{c\}$ and $G_C = (\{\mathbf{X}_c\}, \mathbf{E})$. Typically, DiGress consists of two components: forward noising process and reverse denoising process. For $t \geq 1$, the forward noising process of DiGress is defined by $q(G^{(t)}|G^{(t-1)})$ and $q(G^{(T)}|G^{(0)}) = \prod_{t=1}^{T} q(G^{(t)}|G^{(t-1)})$.

For the reverse denoising process, given $G^{(t)}$, DiGress predicts the clean graph $G^{(0)}$ by a denoising neural network $\phi_\theta$ (parameterized by $\theta$) and obtains the reverse denoising process $p_\theta$ as follows:

$$p_\theta(G^{(t-1)}|G^{(t)}) = q(G^{(t-1)}|G^{(t)}, G^{(0)})p_\theta(G^{(0)}|G^{(t)});$$
$$q(G^{(t-1)}|G^{(t)}) \propto q(G^{(t)}|G^{(t-1)})q(G^{(t-1)}|G^{(0)}), \tag{1}$$

in which $q(G^{(t-1)}|G^{(t)})$ can be approximated by the noising process. To enable single conditional graph generation, DiGress guides the reverse denoising process by a machine learning model $f$ (i.e., a regression model) to push the predicted distribution toward graphs fulfilling the condition $c$. $f$ is trained to predict the condition $c$ of the input graph $G$ from the noised version $G^{(t)}$ such that $c \approx \hat{c} = f(G^{(t)})$. The reverse denoising process guided by a single condition is presented as follows:

$$q(G^{(t-1)}|G^{(t)}, c) \propto q(c|G^{(t-1)})q(G^{(t-1)}|G^{(t)}), \tag{2}$$

where the first term is approximated by the learned distribution of the regression model, and the second term is approximated by the unconditional diffusion model.

## 3.3 DUAL CONDITIONAL GRAPH SYNTHESIZING

Here, we present the overall learning framework of `CDGraph` and its novel features for integrating dual conditions. Specifically, to capture the dependency between two exogenous conditions, `CDGraph` introduces *co-evolution dependency* to model the co-evolving diffusion process of dual conditions in the reverse denoising process with the notions of *social homophily* and *social contagion* so that the dependencies between node conditions and edge connections can be captured in the diffusion process. Then, to further fulfill dual conditions, we design a *dual condition classifier* to guide the co-evolution diffusion process, modulating the estimated distribution to align with graphs satisfying specified conditions by the classifier loss.

Specifically, `CDGraph` comprises the forward noising process and the reverse denoising process, which are illustrated in Figure 2. Given $C = \{c_1, c_2\}$ and the condition indication graph

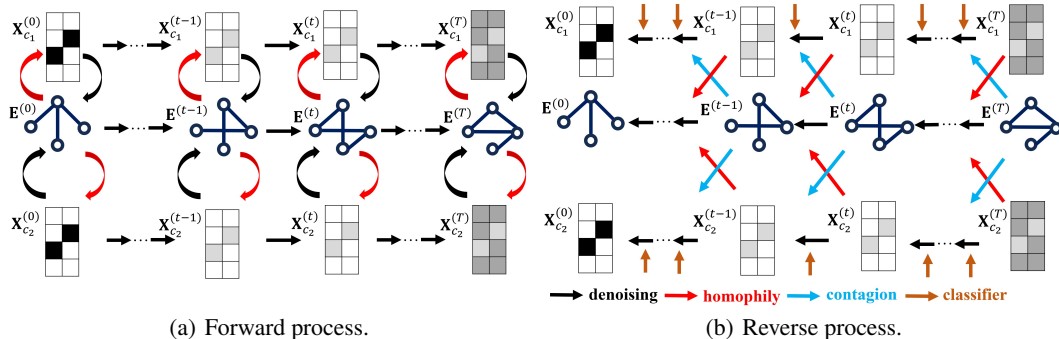

(a) Forward process.            (b) Reverse process.

Figure 2: Workflow of CDGraph, where $\mathbf{E}$ denotes the edge embedding, and $\mathbf{X}_{c_1}$, and $\mathbf{X}_{c_2}$ are embeddings of nodes indicating the satisfaction of conditions $c_1$ and $c_2$.

$G_C = (\{\mathbf{X}_{c_1}, \mathbf{X}_{c_2}\}, \mathbf{E})$, we model the forward noising processes for each of the specified conditions $c_i, \forall i \in \{1, 2\}$ as follows:

$$q(\mathbf{X}_{c_i}^{(t)}|\mathbf{X}_{c_i}^{(t-1)}) = \mathbf{X}_{c_i}^{(t-1)}\mathbf{Q}_{X_{c_i}}^{(t)}; q(\mathbf{E}^{(t)}|\mathbf{E}^{(t-1)}) = \mathbf{E}^{(t-1)}\mathbf{Q}_E^{(t)},$$

where $\mathbf{Q}_{X_{c_i}}^{(t)} \in \mathbb{R}^{2 \times 2}$ and $\mathbf{Q}_E^{(t)} \in \mathbb{R}^{2 \times 2}$ are transition matrices for $\mathbf{X}_{c_i}$ and $\mathbf{E}$, respectively. Then we can show that $q(\mathbf{X}_{c_i}^{(t)}|\mathbf{X}_{c_i}^{(t-1)})$ and $q(\mathbf{E}^{(t)}|\mathbf{E}^{(t-1)})$ obey Bernoulli distributions as follows:

$$q(\mathbf{X}_{c_i}^{(t)}|\mathbf{X}_{c_i}^{(t-1)}) = \mathcal{B}(\mathbf{X}_{c_i}^{(t)}; (1 - \beta_t)\mathbf{X}_{c_i}^{(t-1)} + \beta_t \mathbf{1}/2); \tag{3}$$

$$q(\mathbf{E}^{(t)}|\mathbf{E}^{(t-1)}) = \mathcal{B}(\mathbf{E}^{(t)}; (1 - \beta_t)\mathbf{E}^{(t-1)} + \beta_t \mathbf{1}/2). \tag{4}$$

The detailed derivations regarding $\mathbf{Q}_{X_{c_i}}^{(t)}$ and $\mathbf{Q}_E^{(t)}$ are provided in Appendix A (Anonymous, 2023).

By considering nodes with two conditions and the dependency between nodes and edges, the overall forward processes of $\mathbf{X}_{c_i}$ and $\mathbf{E}$ are formulated as follows:

$$q(\mathbf{X}_{c_i}^{(0:T)}) = \prod_{t=1}^{T} q(\mathbf{X}_{c_i}^{(t)}|\mathbf{X}_{c_i}^{(t-1)}, \mathbf{E}^{(t-1)})q(\mathbf{E}^{(t-1)}|\mathbf{X}_{c_i}^{(t-1)}),$$

$$q(\mathbf{E}^{(0:T)}) = \prod_{t=1}^{T} q(\mathbf{E}^{(t)}|\mathbf{E}^{(t-1)}, \mathbf{X}_{c_i}^{(t-1)})q(\mathbf{X}_{c_i}^{(t-1)}|\mathbf{E}^{(t-1)}). \tag{5}$$

The overall forward noising processes of $\mathbf{X}_{c_1}^{(t)}$, $\mathbf{E}^{(t)}$, and $\mathbf{X}_{c_2}^{(t)}$ are illustrated in Fig. 2(a) from above to below, and each $\mathbf{X}_{c_i}^{(t)}(i = 1, 2)$ has dependency with $\mathbf{E}^{(t)}$. The red bending arrow represents the conditional probability distribution $q(\mathbf{X}^{(t)}|\mathbf{E}^{(t)})$ of $\mathbf{X}^{(t)}$ given $\mathbf{E}^{(t)}$, and the black bending arrow represents the conditional probability distribution $q(\mathbf{E}^{(t)}|\mathbf{X}^{(t)})$ of $\mathbf{E}^{(t)}$ given $\mathbf{X}^{(t)}$. The detailed forward transition distribution with dependency in the above processes is defined and analyzed in Appendix A (Anonymous, 2023).

### 3.3.1 CO-EVOLUTION DEPENDENCY

Different from DiGress, CDGraph exploits *co-evolution dependency* to model the intricate dependency across conditions of the nodes and connections between them in the denoising process, which is detailed in Appendix A (Anonymous, 2023). This emphasis on capturing the relationship between $\mathbf{X}_{c_i}^{(t)}, i \in \{1, 2\}$ and $\mathbf{E}^{(t)}$ are crucial for the precise reconstruction of the input graphs. To capture the dependencies between the connection between nodes and condition satisfaction of the nodes in a social graph, we build a denoising model incorporating two phenomena in social networks: *social homophily* and *social contagion*. The concept of the above denoising process is illustrated in the lower half of Figure 2(b). The reverse denoising processes of $\mathbf{X}_{c_1}^{(t)}$, $\mathbf{E}^{(t)}$, and $\mathbf{X}_{c_2}^{(t)}$ are illustrated in Fig. 2(b) from above to below, and the blue arrow represents the co-evolution based on social contagion; the red arrow represents the co-evolution based on social homophily.

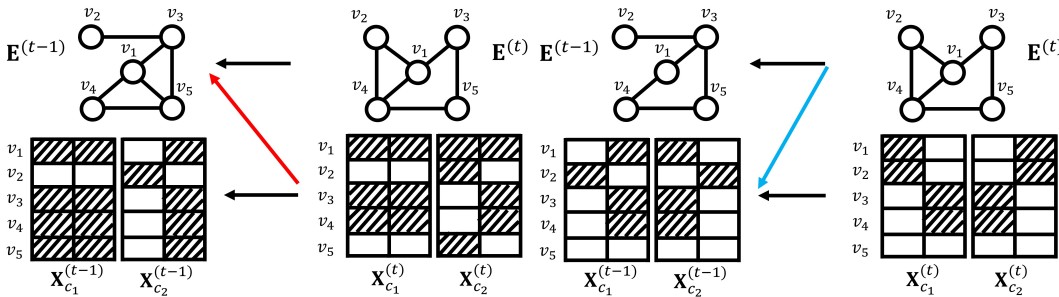

(a) Social Homophily-based Co-evolution.  (b) Social Contagion-based Co-evolution.

Figure 3: Co-evolution dependency incorporating social homophily and social contagion.

The notions of *social homophily* and *social contagion* in the diffusion process in `CDGraph` are illustrated in Figure 3. In Fig. 3(a), as denoising from $G_C^{(t)}$ to $G_C^{(t-1)}$, the nodes with similar attributes $(v_1, v_3, v_4)$ in $G_C^{(t-1)}$ tend to have links between them. In Fig. 3(b), as denoising from $G_C^{(t)}$ to $G_C^{(t-1)}$, the edges incident to nodes with similar attributes $(v_3, v_4)$ tend to cause the other node $(v_1)$ have the same attribute.

Assuming that $\mathbf{X}_{c_i}^{(t-1)}$, $\mathbf{X}_{c_j}^{(t-1)}$ and $\mathbf{E}^{(t-1)}$ are conditionally independent given $\mathbf{X}_{c_i}^{(t)}$, $\mathbf{X}_{c_j}^{(t)}$ and $\mathbf{E}^{(t)}$, the reverse denoising process can be further decomposed as follows:

$$p_\theta(\mathbf{X}_{c_i}^{(t-1)}, \mathbf{X}_{c_j}^{(t-1)}, \mathbf{E}^{(t-1)}|\mathbf{X}_{c_i}^{(t)}, \mathbf{X}_{c_j}^{(t)}, \mathbf{E}^{(t)})$$
$$= p_\theta(\mathbf{X}_{c_i}^{(t-1)}|\mathbf{X}_{c_i}^{(t)}, \mathbf{E}^{(t)})p_\theta(\mathbf{X}_{c_j}^{(t-1)}|\mathbf{X}_{c_j}^{(t)}, \mathbf{E}^{(t)})p_\theta(\mathbf{E}^{(t-1)}|\mathbf{X}_{c_i}^{(t)}, \mathbf{E}^{(t)}, \mathbf{X}_{c_j}^{(t)}), \qquad (6)$$

in which the first two terms represent *social contagion* that can be used to denoise node conditions from given edges, and the third term represents *social homophily* that can be exploited to denoise connections between nodes from given conditions of nodes. Note that $\mathbf{X}_{c_i}^{(t-1)}$ is independent on $\mathbf{X}_{c_j}^{(t)}$ (and $\mathbf{X}_{c_j}^{(t-1)}$ is independent on $\mathbf{X}_{c_i}^{(t)}$) for distinct $c_i$ and $c_j$.

**Social Homophily-based Co-evolution.**  In this paragraph, we discuss how to guide the diffusion process with social homophily, which states that nodes with similar conditions tend to have edges between them. The social homophily-based co-evolution aims to denoise edge embeddings from given node embeddings of each condition (as stated in Eq. 7). Accordingly, we consider the following denoising process:

$$p_\theta(\mathbf{E}^{(0:T)}) = p_\theta(\mathbf{X}_{c_i}^{(T)}) \prod_{t=1}^{T} p_\theta(\mathbf{E}^{(t-1)}|\mathbf{E}^{(t)}, \mathbf{X}_{c_i}^{(t)});$$
$$p_\theta(\mathbf{E}^{(t-1)}|\mathbf{E}^{(t)}, \mathbf{X}_{c_i}^{(t)}) = \mathcal{B}(\mathbf{p}_\theta^{(homo)}), \qquad (7)$$

where

$$\mathbf{p}_\theta^{(homo)} = \sum_{\hat{\mathbf{E}}^{(0)} \in \{0,1\}} q(\mathbf{E}^{(t-1)}|\mathbf{E}^{(t)}, \hat{\mathbf{E}}^{(0)})\hat{p}_e(\hat{\mathbf{E}}^{(0)}|\mathbf{E}^{(t)}, \mathbf{X}_{c_i}^{(t)}),$$

and $\hat{p}_e$ is the distribution learned to predict $\mathbf{E}^{(0)}$ from $\mathbf{E}^{(t)}$ conditioned on $\mathbf{X}_{c_i}^{(t)}, \mathbf{X}_{c_i}^{(t)}$ by denoising network $\phi_\theta$.

The loss function of social homophily-based co-evolving diffusion can be derived as follows:

$$\mathcal{L}_{homo} = \sum_{t=2}^{T-1} D_{KL}[q(\mathbf{E}^{(t-1)}|\mathbf{E}^{(t)}, \mathbf{E}^{(0)})\|p_\theta(\mathbf{E}^{(t-1)}|\mathbf{E}^{(t)}, \mathbf{X}_{c_i}^{(t)}, \mathbf{X}_{c_j}^{(t)})]$$
$$+ D_{KL}[q(\mathbf{E}^{(T)}|\mathbf{E}^{(0)})\|p(\mathbf{E}^{(T)})] - \log p_\theta(\mathbf{E}^{(0)}|\mathbf{E}^{(1)}, \mathbf{X}_{c_i}^{(1)}, \mathbf{X}_{c_j}^{(1)}), \qquad (8)$$

where the first term is the loss for diffusion process; the second term is the loss for prior distribution; the third term is the loss for reconstruction.

**Social Contagion-based Co-evolution.** In this paragraph, we discuss how to guide the diffusion process with social contagion, which states that nodes connected with edges tend to have similar conditions. The social contagion-based co-evolution aims to denoise node embeddings of each condition from given edge embeddings (as stated in Eq. 9). Accordingly, we consider the following denoising process:

$$p_\theta(\mathbf{X}_{c_i}^{(0:T)}) = p_\theta(\mathbf{E}^{(T)}) \prod_{t=1}^{T} p_\theta(\mathbf{X}_{c_i}^{(t-1)}|\mathbf{X}_{c_i}^{(t)}, \mathbf{E}^{(t)});$$

$$p_\theta(\mathbf{X}_{c_i}^{(t-1)}|\mathbf{X}_{c_i}^{(t)}, \mathbf{E}^{(t)}) = \mathcal{B}(\mathbf{p}_\theta^{(cont)}), \qquad (9)$$

where

$$\mathbf{p}_\theta^{(cont)} = \sum_{\hat{\mathbf{X}}_{c_i}} q(\mathbf{X}_{c_i}^{(t-1)}|\mathbf{X}_{c_i}^{(t)}, \hat{\mathbf{X}}_{c_i}^{(0)}) \hat{p}_{c_i}(\hat{\mathbf{X}}_{c_i}^{(0)}|\mathbf{X}_{c_i}^{(t)}, \mathbf{E}^{(t)}),$$

and $\hat{p}_{c_i}$ is the distribution learned to predict $\mathbf{X}_{c_i}^{(0)}$ from $\mathbf{X}_{c_i}^{(t)}$ conditioned on $\mathbf{E}^{(t)}$ by denoising network $\phi_\theta$.

And the loss function for social contagion-based co-evolution can be derived as follows:

$$\mathcal{L}_{cont} = \sum_{c_i \in C} \sum_{t=2}^{T-1} D_{KL}[q(\mathbf{X}_{c_i}^{(t-1)}|\mathbf{X}_{c_i}^{(t)}, \mathbf{X}_{c_i}^{(0)})\|p_\theta(\mathbf{X}_{c_i}^{(t-1)}|\mathbf{X}_{c_i}^{(t)}, \mathbf{E}^{(t)})]$$

$$+ D_{KL}[q(\mathbf{X}_{c_i}^{(T)}|\mathbf{X}_{c_i}^{(0)})\|p(\mathbf{X}_{c_i}^{(T)})] - \log p_\theta(\mathbf{X}_{c_i}^{(0)}|\mathbf{X}_{c_i}^{(1)}, \mathbf{E}^{(1)}). \qquad (10)$$

The overall loss function of the co-evolution diffusion process is $\mathcal{L} = \mathcal{L}_{homo} + \mathcal{L}_{cont}$, which jointly optimizes the discrepancy in diffusion process and graph reconstruction in order to synthesize graphs with properties of social homophily and social contagion. The pseudocode of training and sampling is depicted in Appendix B (Anonymous, 2023).

### 3.3.2 DUAL-CONDITION CLASSIFIER

Afterward, `CDGraph` leverages *dual conditional classifier* to enable joint guidance for $p_\theta(G_C^{(t-1)}|G_C^{(t)})$ to fulfill dual conditions, instead of single conditional guidance in DiGress. To guide the diffusion process with two specified conditions jointly, we exploit the concept of conditional guidance and model the guidance distribution for the specified conditions $c_i$ and $c_j$ ($i \neq j$) with a classifier such that the generated graphs are classified according to whether a majority of the nodes satisfy both of the specified conditions or not. Note that the diffusion process satisfies the Markovian property: $q(G_C^{(t-1)}|G_C^{(t)}, c_i) = q(G_C^{(t-1)}|G_C^{(t)}), \forall i$. The conditional diffusion model exploits a pre-trained classifier to guide the learning process, ensuring that the model learns the specified conditions by classifying graphs according to whether the majority of nodes in a graph satisfy both conditions. Specifically, a pre-trained classifier $f$ learns the distribution $p(c_i|G^{(t)})$ for a graph classification problem $c_i \approx f(G^{(t)})$ in each step $t$ of the denoising process (as shown by the brown arrow in Fig. 2(b)) so that the denoised graphs can be predicted to have the specified condition $c_i$, which can be derived by Bayesian Theorem. As there is one additional condition $c_j$ to be considered, we need to consider a conditional probability distribution $p(c_j|G^{(t)}, c_i)$ to further guide the denoising process to meet the second condition given that the first condition is satisfied. Thus, from the results derived in the single-conditional denoising process, the co-evolution reverse denoising process with dual-condition classifier can be derived as follows:

$$q(G_C^{(t-1)}|G_C^{(t)}, c_i, c_j) = \frac{q(c_i|G_C^{(t-1)}, G_C^{(t)}, c_j)q(G_C^{(t-1)}, G_C^{(t)}, c_j)}{q(c_i|G_C^{(t)}, c_j)q(c_j|G_C^{(t)})q(G_C^{(t)})}$$

$$= \frac{q(c_i|G_C^{(t-1)}, G_C^{(t)}, c_j)q(G_C^{(t-1)}|G_C^{(t)}, c_j)}{q(c_i|G_C^{(t)}, c_j)} \propto q(c_i|G_C^{(t-1)}, c_j)q(c_j|G_C^{(t-1)})q(G_C^{(t-1)}|G_C^{(t)}),$$

where the first two terms enable the hierarchical guidance of the conditions to guide the denoised graph $G_C^{(t-1)}$ satisfying one condition $c_j$ (i.e., more than half of the nodes in $G_C^{(t-1)}$ satisfy condition

Table 1: Validity and discrepancy with the input graphs of different approaches on all datasets.

| Dataset | Facebook | | | | | BlogCatalog | | | | |
|---------|----------|------|------|---------|--------------|----------|------|------|---------|--------------|
| Metric | Validity | Node | Edge | Density | Clust. coeff. | Validity | Node | Edge | Density | Clust. coeff. |
| RW | 0 | 5.170 | 8.225 | 0.698 | 0.242 | - | - | - | - | - |
| SPECTRE | 0.448 | 0.243 | 1.795 | 1.119 | 0.313 | 0.3 | 0.026 | 6.081 | 5.494 | 0.098 |
| GSM | 0.272 | **0.100** | 0.551 | 0.392 | 0.031 | 0.29 | **0.013** | 3.486 | 3.441 | 0.063 |
| EDGE | 0.262 | 0.736 | 1.983 | 0.815 | 0.028 | 0.292 | 0.414 | 0.510 | 0.256 | **0.014** |
| DiGress | 0.375 | 0.111 | 0.852 | 0.457 | 0.143 | 0.3125 | 0.164 | 0.413 | 0.699 | 0.266 |
| CDGraph | **1** | 0.149 | **0.186** | **0.002** | **0.022** | **1** | 0.020 | **0.259** | **0.219** | 0.031 |

| Dataset | Twitter | | | | | Flickr | | | | |
|---------|---------|------|------|---------|--------------|--------|------|------|---------|--------------|
| Metric | Validity | Node | Edge | Density | Clust. coeff. | Validity | Node | Edge | Density | Clust. coeff. |
| RW | - | - | - | - | - | - | - | - | - | - |
| SPECTRE | 0.239 | 0.163 | 1.061 | 0.441 | 0.359 | 0.323 | 0.053 | 2.115 | 2.275 | 1.006 |
| GSM | 0.309 | 0.205 | 0.417 | 0.458 | 0.011 | 0.269 | **0.005** | 1.055 | 1.039 | **0.014** |
| EDGE | 0.291 | 0.101 | 0.034 | **0.053** | **0.002** | 0.286 | 0.439 | 0.809 | 0.695 | **0.014** |
| DiGress | 0.5 | **0.029** | 0.301 | 0.084 | 0.174 | 0.375 | 0.090 | 0.233 | 0.139 | 0.286 |
| CDGraph | **1** | 0.045 | **0.001** | 0.201 | 0.016 | **1** | 0.008 | **0.056** | **0.068** | 0.068 |

$c_j$), and then satisfying the other condition $c_i$ (i.e., more than half of the nodes in $G_C^{(t-1)}$ satisfy condition $c_i$) given that $G_C^{(t-1)}$ satisfies condition $c_j$. The third term $q(G_C^{(t-1)}|G_C^{(t)})$ is approximated by the aforementioned co-evolution denoising process with *social homophily* and *social contagion*. Note that, compared to DiGress and other previous studies, the proposed guidance model has more capability to guide the denoising process such that the denoised graphs meet both of the specified conditions if the specified conditions have a negative correlation. The pseudocode of conditional sampling is depicted in Appendix B (Anonymous, 2023).

# 4 EMPIRICAL EVALUATION

## 4.1 SETUP

The details of the experimental setup are presented in Appendix G (Anonymous, 2023).

**Datasets**. We conduct the experiments on four real-world social graphs: Facebook, Twitter, Flickr, and BlogCatalog. We sample ego networks with at maximum 100 nodes for each dataset.

**Baselines**. We compare CDGraph with five baselines: RW (Nakajima & Shudo, 2022), SPEC-TRE (Martinkus et al., 2022), GSM (Niu et al., 2020), EDGE (Chen et al., 2023), and DiGress (Vignac et al., 2023).

**Metrics**. Our evaluation considers three aspects: 1) Validity (the higher, the better); 2) discrepancy with the input graphs (the lower, the better), including the relative error ratios of #nodes, #edges, and density, as well as the maximum mean discrepancy (MMD) of clustering coefficients; 3) homophily discrepancy with the input graphs (the lower, the better), including the relative error ratios of assortativity and EI homophily index.

## 4.2 PERFORMANCE IN DUAL CONDITIONAL SOCIAL GRAPH GENERATION

Table 1 presents the validity and discrepancy with the input graphs of different approaches on all datasets when the dual conditions are weakly positively correlated.[1] Additional cases of dual conditions and visualizations are presented in Appendix C (Anonymous, 2023) due to space constraints.

**Validity.** Compared with the baselines, CDGraph is demonstrated to be able to generate graphs with the majority of nodes satisfying both of the specified conditions since it exploits the guidance of the dual condition classifier. Among the baselines, DiGress usually achieves the highest validity since it is aware of dual conditions, whereas the other baselines do not account for them. Nevertheless, DiGress neglects the dependency between dual conditions and only achieves half of the validity achieved by CDGraph.

**Discrepancy with the input graphs.** CDGraph has superior performance in the relative error ratios of the edge number and density on Facebook and BlogCatalog. The relative error ratio of density in EDGE and DiGress is lower than in CDGraph since EDGE focuses on guiding degree changes, and DiGress attempts to preserve graph sparsity in the diffusion process. Nevertheless,

---

[1]The results of RW on BlogCatalog, Twitter, and Flickr are omitted due to the out-of-memory issue.

Table 2: Homophily discrepancy with the input graphs of different approaches on Facebook.

|  | Assortativity | EI homophily index |
|---|---|---|
| RW | 0.351 | 1.437 |
| SPECTRE | 0.042 | 0.172 |
| GSM | 0.043 | 0.175 |
| EDGE | **0.006** | **0.025** |
| DiGress | 0.067 | 0.275 |
| CDGraph | 0.013 | 0.052 |

(a) Input graph.    (b) Generation of SPECTRE.    (c) Generation of GSM.

(d) Generation of EDGE.    (e) Generation of DiGress.    (f) Generation of `CDGraph`

Figure 4: Visualization results of Twitter.

density refers to the ratio of the number of edges to the maximum possible number of edges (which is related to the square of the number of nodes) in a graph. Hence, it is more comprehensive to simultaneously examine the relative error ratios of nodes, edges, and density. In `CDGraph`, the relative error ratios for both nodes and edges are less than 0.05. Overall, the structure of the graphs generated by `CDGraph` remains very close to that of the input graphs. Regarding the MMD of clustering coefficients, `CDGraph` demonstrates the best performance on Facebook and also ranks second on Flickr and BlogCatalog. As the clustering coefficient is a fundamental property of social graphs, the results highlight that `CDGraph` surpasses the baselines in generating social graphs.

**Homophily discrepancy with the input graphs.** Table 2 presents homophily discrepancy with the input graphs on Facebook. For both assortativity and the EI homophily index, `CDGraph` obtains the second-lowest relative error ratios, indicating the effectiveness of the design based on social homophily and social contagion co-evolution. It is worth noting that EDGE achieves the lowest relative error ratios, as its generation process is guided by nodes' degrees, which may implicitly learn associations with node attributes from the input graphs. Other baselines, lacking specific consideration for the relationship between structure and attributes, exhibit higher relative error ratios.

**Visualization.** Figure 8 visualizes the input and generated graphs for Twitter. Nodes in blue indicate that both specified conditions are met, while nodes in yellow and green represent those satisfying only one condition. Conversely, nodes in red do not meet any of the conditions. Obviously, the generation results of `CDGraph` are the closest to the input graph, thanks to the denoising process based on social homophily and social contagion.

## 5 CONCLUSION

In this paper, we make the first attempt to develop a conditional diffusion model for social networks, `CDGraph`, which aims to synthesize social graphs satisfying two specified conditions. `CDGraph` introduces a novel feature of *co-evolution dependency* integrating *social homophily* and *social contagion*, allowing `CDGraph` to capture the interdependencies between specified conditions. Moreover, `CDGraph` exploits the *dual-condition classifier* to ensure that discrepancies in the diffusion process and structure reconstruction are jointly optimized. The experimental results manifest that `CDGraph` achieves lower discrepancies in various network statistics compared with the baselines.

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
