# OpenReview forum: "CDGraph: Dual Conditional Social Graph Synthesizing via Diffusion Model"
_ICLR.cc/2024/Conference — Submitted to ICLR 2024_

### Official Review · Reviewer_zZ91 · 2023-10-31

**Soundness:** 2 fair
**Presentation:** 2 fair
**Contribution:** 2 fair
**Rating:** 5
**Confidence:** 3

**Summary:**

This paper proposes a dual conditional diffusion model for social graph synthesizing. The proposed model can capture the dependency between two specified conditions, social homophily and social contagion. The experimental results show the effectiveness of the proposed model.

**Strengths:**

1. A new diffusion model for social graph synthesizing, which is under-exploration.
2. The proposed model can capture the dependency between social homophily and social contagion, which is important to social graphs.
3. The completeness of the work is satisfactory, including the design of the loss and classifier.

**Weaknesses:**

1. The author's real motivation is confusing. Why the dual conditional model is important and why not more (than two) should be clearly pointed out before explaining the challenges (maybe not just giving examples).
2. The reason why the authors choose social homophily and social contagion as the two specific conditions in this paper is not clear. The author's criteria or basis for selecting these two aspects should be introduced. No explanation of the above two concepts is not provided when they first appear.
3. Both social homophily and social contagion are based on the assumption of network homogeneity. However, more and more works point out the heterogeneity of real-world networks, especially in social networks (e.g., Facebook and Twitter). Therefore, it is concerning whether the assumptions on which the method is based are good for social graphs.
4. The scalability of the proposed method is slightly unsatisfactory. As a dual conditional method, the proposed method uses the conditions between the nodes/edges to implement two specific conditional constraints through the transfer of point and edge information. It is not explored whether the model can be applied to any two constraints and how it should be extended if both conditions are at the node level (independent of edges).
5. Figure 2 should be improved for more clarity with some explanations either in the figure or in the caption.
6. The ablation studies of each condition and loss are missing.

**Questions:**

Please refer to Weaknesses.

---

> ### Author Response · Authors · 2023-11-21
> **Answers to Questions 1~2**
>
> **Q1 and Q2**: The motivation and importance of a dual conditional diffusion model, the reason why not more than two conditions, and the explanation of social homophily and social contagion.
>
> **Response**:  The generation of dual conditional social graphs is highly valuable for social applications, as these applications often require the consideration of users' diverse social contexts and connections, which typically satisfy more than one condition in real-world applications. Moreover, they should also retain the unique characteristics of social graphs, i.e., social homophily and social contagion, to ensure that the phenomena inherent in social applications can be captured. Social homophily [6], which involves the tendency to form social connections with people who share similar profiles, plays a significant role as a primary factor in link formation [1]. On the other hand, social contagion [7,8], where behavior spontaneously spreads through a social graph, is largely influenced by network topology [2]. Since homophily and contagion are generally intertwined [3], many studies often simultaneously consider both to facilitate social applications. For instance, Li et al. [4] explore the effects of social homophily and contagion on users' behavior for social recommendation. Sankar et al. [5] leverage social homophily and contagion to accurately predict the spread of information.
>
> In downstream applications like social marketing, it is common to select individuals with strong influence within their social circles for word-of-mouth promotion. These social circles are parts of users' ego social graphs, preserving the essential characteristics of social homophily and social contagion to accurately capture social influence propagation. Specifically, for luxury golf club brands targeting golf enthusiasts and high-income individuals, analyzing users' influence within social circles that include friends sharing a passion for golf and possessing high incomes can help identify individuals well-suited for promoting the brands. Similarly, in link prediction, recommending friends with similar social circles is fundamental. Therefore, focusing on users' social circles characterized by a shared interest in birdwatching and frequent visits to locations in Japan ensures precise recommendations of friends involved in birdwatching within Japan.
>
> Additionally, we agree that a multi-conditional model is expected to be more general and applicable to a broader range of social applications. We provide the method of extending CDGraph to support additional conditions as follows; however, due to time constraints, the implementation of such extensions is deferred to future work.
>
> To support more than two conditions, we need to modify the probability distribution of the reverse process by augmenting the node embeddings $\mathbf{X_{c_i}}$ to indicate the satisfaction of the conditions. And the probability distribution of the classification guidance model can be modified by additionally considering the dependency between the conditions.
> We noticed that Eq. (6) can be extended by adding more feature matrices of node conditions, which can be written as follows.
> $$
> p_{\theta} ( \lbrace \mathbf{X}_{c_i}^{(t-1)} \rbrace _{i=1}^{K}, \mathbf{E}^{(t-1)} | \lbrace \mathbf{X} _{c_i}^{(t-1)} \rbrace _{i=1}^{K}, \mathbf{E}^{(t)}) = \prod _{i=1}^{K} p _{\theta} (\mathbf{X} _{c_i}^{(t-1)} | \mathbf{X} _{c_i}^{(t)}, \mathbf{E}^{(t)}) p _{\theta} (\mathbf{E}^{(t-1)} | \mathbf{E}^{(t)}, \lbrace \mathbf{X} _{c_i}^{(t-1)} \rbrace _{i=1}^{K}),
> $$
>
> where $K$ is the number of specified conditions.
> On the other hand, the dual-conditional classification guidance model can also be extended by specifying more conditions as follows:
> $$
> q (G_C^{(t-1)} | G_C^{(t)}, \lbrace c_i \rbrace _{i=1}^{K}) \propto q(c_i | G_C^{(t-1)}, \lbrace c_j \rbrace _{j \neq i}) \prod _{j \neq i} q (c_j | G_C^{(t-1)}) q (G_C^{(t-1)} | G_C^{(t)})
> $$
>
> where $K$ is the number of specified conditions, and $\lbrace c_j \rbrace _{j \neq i}$ are conditions dependent with $c_i$.

---

> > ### Author Response · Authors · 2023-11-21
> > **References**
> >
> > [1] E. Lee, F. Karimi, C. Wagner, H.-H. Jo, M. Strohmaier, and M. Galesic. Homophily and minority-group size explain perception biases in social networks. Nature human behaviour, 2019.
> >
> > [2] N. Horsevad, D. Mateo, R. E. Kooij, A. Barrat, and R. Bouffanais. Transition from simple to complex contagion in collective decision-making. Nature Communications, 2022.
> >
> > [3] C.R. Shalizi and A. C. Thomas. Homophily and contagion are generically confounded in observational social network studies. Sociological Methods & Research, 2011.
> >
> > [4] N. Li, C. Gao, D. Jin, and Q. Liao. Disentangled modeling of social homophily and influence for social recommendation. IEEE Transactions on Knowledge and Data Engineering, 2023.
> >
> > [5] A. Sankar, X. Zhang, A. Krishnan, and J. Han. Inf-VAE: A variational autoencoder framework to integrate homophily and influence in diffusion prediction. In ACM WSDM, 2020.
> >
> > [6] H. Bisgin, N. Agarwal & X. Xu, A study of homophily on social media. World Wide Web 15, 213–232 (2012).
> >
> > [7] Marios Papachristou, Siddhartha Banerjee, and Jon Kleinberg. 2023. Dynamic Interventions for Networked Contagions. In Proceedings of the ACM Web Conference 2023 (WWW '23). Association for Computing Machinery, New York, NY, USA, 3519–3529.
> >
> > [8] Julie Jiang, Ron Dotsch, Mireia Triguero Roura, Yozen Liu, Vítor Silva, Maarten W. Bos, and Francesco Barbieri. 2023. Reciprocity, Homophily, and Social Network Effects in Pictorial Communication: A Case Study of Bitmoji Stickers. In CHI '23.

---

> ### Author Response · Authors · 2023-11-21
> **Answers to Questions 3~6**
>
> **Q3**: The assumptions of network homogeneity/heterogeneity.
>
> **Response**: Network homogeneity (or heterogeneity) generally refers to the degree of similarity (or dissimilarity) among users in a social graph. Social homophily and social contagion are widespread phenomena on social graphs, irrespective of whether the graph exhibits homogeneity or heterogeneity. Nevertheless, we agree with the reviewer that network heterogeneity is observed in real-world social networks, indicating that users can also become friends with others with diverse profiles. It is important to note that CDGraph is currently in its early stages and is primarily focusing on fundamental social homophily and social contagion. In the future, we can aim to support a more diverse range of social phenomena by modifying the co-evolution dependency module and introducing an additional distribution to capture the heterogeneity.
>
> **Q4**: Scalability of the proposed method to any two constraints and how to extend if both conditions are at the node level.
>
> **Response**: We agree with the reviewer regarding the significance of scalability in accommodating any two conditions at various levels, whether at the node, edge, or a combination of both. In the context of this study, CDGraph is designed to produce dual conditional social graphs, with both conditions applied at the node level. In our current work, CDGraph focuses on generating dual conditional social graphs, with both conditions applied at the node level. Acknowledging the importance of the reviewer's suggestion, we plan to extend the dual conditional classifier module to provide flexibility in supporting conditions at any level. This improvement will be incorporated into the final manuscript.
>
> **Q5**: Explanations regarding Figure 2 (in the original paper) with a detailed explanation of the workflow of CDGraph.
>
> **Response**:  The workflow of CDGraph is updated in Figure 2 of the revised paper (previously Figure 3 in the original paper), and we provide the detailed explanation as follows: The forward noising processes of $\mathbf{X_{c_{1}}}^{(t)}$, $\mathbf{E}^{(t)}$, and $\mathbf{X_{c_{2}}}^{(t)}$ are illustrated from above to below, and each $\mathbf{X_{c_{i}}}^{(t)} (i=1,2)$ has dependency with $\mathbf{E}^{(t)}$. The red bending arrow represents the conditional probability distribution $q(\mathbf{X_{c_{i}}}^{(t)}|\mathbf{E}^{(t)})$ of $\mathbf{X_{c_{i}}}^{(t)}$ given $\mathbf{E}^{(t)}$, and the black bending arrow represents the conditional probability distribution $q(\mathbf{E}^{(t)}|\mathbf{X_{c_{i}}}^{(t)})$ of $\mathbf{E}^{(t)}$ given $\mathbf{X_{c_{i}}}^{(t)}$.
>
> On the other hand, the reverse denoising processes of $\mathbf{X_{c_{1}}}^{(t)}$, $\mathbf{E}^{(t)}$, and $\mathbf{X_{c_{2}}}^{(t)}$ are illustrated from above to below, and the blue arrow represents the co-evolution based on social contagion; the red arrow represents the co-evolution based on social homophily.
>
> The illustrations of social homophily and social contagion are updated in Figure 3 of the revised paper (previously Figure 2 in the original paper), and we provide a detailed explanation as follows: The social homophily-based coevolution aims to denoise edge embeddings from given node embeddings of each condition (as stated in Eq. (7)); the social contagion-based co-evolution aims to denoise node embeddings of each condition from given edge embeddings (as stated in Eq. (9)).
>
> **Q6**: Ablation study.
>
> **Response**: We apologize that the results of the ablation study could not be produced within the time limit because the interdependence among modules is complex, and removing the implementation of each module requires a bit more time. We plan to introduce three variants of CDGraph: CDGraph-coevolve, CDGraph-class, and CDGraph-plain. CDGraph-coevolve removes the co-evolution dependency module in the reverse denoising process, CDGraph-class discards the dual conditional classifier module in the reverse denoising process, while CDGraph-plain omits both modules. The ablation study will be included in the final manuscript.

---

### Official Review · Reviewer_QDkE · 2023-11-01

**Soundness:** 2 fair
**Presentation:** 2 fair
**Contribution:** 2 fair
**Rating:** 5
**Confidence:** 4

**Summary:**

This paper proposes a new model, CDGraph, for generating social networks based on specified conditions. The model incorporates social homophily and contagion to maintain connectivity between nodes while satisfying the conditions. It also introduces a novel classifier loss to guide the training process. CDGraph outperforms four existing methods in terms of dual-conditional validity and social network metrics.

**Strengths:**

1. The authors of this paper have developed a novel dual-conditional graph diffusion model called CDGraph for synthesizing social graphs.
2. They have introduced a co-evolution dependency feature that incorporates social homophily and social contagion, allowing for the preservation of structural information between nodes that satisfy specified conditions.
3. Additionally, they have proposed a unique loss function for the dual-condition classifier, which guides the denoising process of CDGraph to optimize both the discrepancy in the diffusion process and the fulfillment of conditions. Through evaluations on four real-world social networks, the authors have demonstrated that CDGraph outperforms existing methods in generating social graphs that meet the specified dual conditions while maintaining important social network properties.

**Weaknesses:**

1.	Contributions only focus on the design of the model, and its motivation is insufficient, and the motivation for designing the model is not clearly explained to the reader.
2.	Some of the figures in the paper are difficult to read, due to the fact that too many models are included in the same figure.
4.	In the experimental part, the author focuses on the lack of sufficient comparison algorithms on the performance of different data sets in the designed method, which is not convincing enough. In particular, the comparison algorithm lacks the classical algorithm.
3.	The author proposed three algorithms, which are CDGraph Training for e pseudocode of the sampling of the diffusion process, CDGraph Training for sampling procedure and Conditional Sampling for pseudocode of the sampling of the diffusion process. But no ablation test was offered. How do these algorithms perform and how are they represented?
4.	The table headers of some graphs are confusing, and the reviewer suggests changing the table headers or table lines appropriately.

**Questions:**

1. How does the conditional diffusion model ensure that the model learns the Specified conditions? Please explain in detail.
2. Methods such as DDPM are specifically mentioned in related work, but they are not compared in related experiments. If there is any relevant work, please add or if there is no or the result is not satisfactory, please explain the reason.

---

> ### Author Response · Authors · 2023-11-21
> **Answers to Questions 1~2**
>
> **Q1**: Detailed explanation of learning the specified conditions.
>
> **Response**: The conditional diffusion model exploits a pre-trained classifier to guide the learning process, ensuring that the model learns the specified conditions by classifying graphs according to whether the majority of nodes in a graph satisfy both conditions. Specifically, a pre-trained classifier $f$ learns the distribution $p(c_{i}|G^{(t)})$ for a graph classification problem $c_{i} \approx f(G^{(t)})$ in each step $t$ of the denoising process (as shown by the brown arrow in Fig. 2(b)) so that the denoised graphs can be predicted to have the specified condition $c_{i}$, which can be derived by Bayesian Theorem. As there is one additional condition $c_{j}$ to be considered, we need to consider a conditional probability distribution $p(c_{j}|G^{(t)}, c_{i})$ to further guide the denoising process to meet the second condition given that the first condition is satisfied.
>
> **Q2**: Explanation regarding the comparisons with DDPM.
>
> **Response**: According to [1], there are two types of DDPMs: continuous and discrete. Since continuous DDPMs may hurt the graph sparsity, and the structural information of noisy graphs in the diffusion process may be lost [2], we opt for discrete DDPMs as our baselines (i.e., DiGress and EDGE), rather than continuous DDPMs.
>
> **References**
>
> [1] Wenqi Fan, et al. "Generative diffusion models on graphs: Methods and applications." IJCAI, 2023.
>
> [2] C. Vignac, I. Krawczuk, A. Siraudin, B. Wang, V. Cevher, and P. Frossard, “Digress: Discrete denoising diffusion for graph generation.” ICLR, 2023.

---

> ### Author Response · Authors · 2023-11-21
> **Responses to weaknesses 1~2**
>
> **W1**: Motivation for the model design.
>
> **Response**: The generation of dual conditional social graphs is highly valuable for social applications, as these applications often require the consideration of users' diverse social contexts and connections, which typically satisfy more than one condition in real-world applications. Moreover, they should also retain the unique characteristics of social graphs, i.e., social homophily and social contagion, to ensure that the phenomena inherent in social applications can be captured. Social homophily [9], which involves the tendency to form social connections with people who share similar profiles, plays a significant role as a primary factor in link formation [4]. On the other hand, social contagion [10, 11], where behavior spontaneously spreads through a social graph, is largely influenced by network topology [5]. Since homophily and contagion are generally intertwined [6], many studies often simultaneously consider both to facilitate social applications. For instance, Li et al. [7] explore the effects of social homophily and contagion on users' behavior for social recommendation. Sankar et al. [8] leverage social homophily and contagion to accurately predict the spread of information.
>
> In social applications like social marketing, it is common to select individuals with strong influence within their social circles for word-of-mouth promotion. These social circles are parts of users' ego social graphs, preserving the essential characteristics of social homophily and social contagion to accurately capture social influence propagation. Specifically, for luxury golf club brands targeting golf enthusiasts and high-income individuals, analyzing users' influence within social circles that include friends sharing a passion for golf and possessing high incomes can help identify individuals well-suited for promoting the brands. Similarly, in link prediction, recommending friends with similar social circles is fundamental. Therefore, focusing on users' social circles characterized by a shared interest in birdwatching and frequent visits to locations in Japan ensures precise recommendations of friends involved in birdwatching within Japan.
>
> **W2**: Detailed explanation of the workflow of CDGraph.
>
> **Response**:  The workflow of CDGraph is updated in Figure 2 of the revised paper (previously Figure 3 in the original paper), and we provide the detailed explanation as follows: The forward noising processes of $\mathbf{X_{c_{1}}}^{(t)}$, $\mathbf{E}^{(t)}$, and $\mathbf{X_{c_{2}}}^{(t)}$ are illustrated from above to below, and each $\mathbf{X_{c_{i}}}^{(t)} (i=1,2)$ has dependency with $\mathbf{E}^{(t)}$. The red bending arrow represents the conditional probability distribution $q(\mathbf{X_{c_{i}}}^{(t)}|\mathbf{E}^{(t)})$ of $\mathbf{X_{c_{i}}}^{(t)}$ given $\mathbf{E}^{(t)}$, and the black bending arrow represents the conditional probability distribution $q(\mathbf{E}^{(t)}|\mathbf{X_{c_{i}}}^{(t)})$ of $\mathbf{E}^{(t)}$ given $\mathbf{X_{c_{i}}}^{(t)}$.
>
> On the other hand, the reverse denoising processes of $\mathbf{X_{c_{1}}}^{(t)}$, $\mathbf{E}^{(t)}$, and $\mathbf{X_{c_{2}}}^{(t)}$ are illustrated from above to below, and the blue arrow represents the co-evolution based on social contagion; the red arrow represents the co-evolution based on social homophily.
>
> The illustrations of social homophily and social contagion are updated in Figure 3 of the revised paper (previously Figure 2 in the original paper), and we provide a detailed explanation as follows: The social homophily-based coevolution aims to denoise edge embeddings from given node embeddings of each condition (as stated in Eq. (7)); the social contagion-based co-evolution aims to denoise node embeddings of each condition from given edge embeddings (as stated in Eq. (9)).

---

> ### Author Response · Authors · 2023-11-21
> **Responses to weaknesses 3**
>
> **W3**: More sufficient comparisons (with the classical algorithm).
>
> **Response**: We add a new baseline RW [1], which is a random-walk-based graph generation algorithm. In addition to the original metrics, including validity and discrepancy with the input graphs (i.e., relative error ratios of #node, #edges, and density, as well as MMD of clustering coefficients), we further consider two homophily metrics, including assortativity and EI homophily index, to evaluate discrepancy with the input graphs. Note that both assortativity and the EI homophily index are presented using relative error ratios. Assume that there are two attributes associated with each node, and each node has a binary value (0/1) for each attribute. A homo-edge is an edge where the end nodes share the same value for every attribute, whereas a hetero-edge is an edge where the end nodes differ in values for at least one attribute.
> * Assortativity [2]: The ratio of the number of homo-edges $|E_{homo}|$ to the number of edges $|E|$ in the whole graph:  $$r=\frac{|E_{homo}|}{|E|}.$$
>
> * EI homophily index [3]: The ratio of the difference between the number of hetero-edges $|E_{hetero}|$ and the number of homo-edges $|E_{homo}|$ to to the number of edges $|E|$ in the whole graph:
> \begin{equation*}
> EI=\frac{|E_{hetero}|-|E_{homo}|}{|E|}.
> \end{equation*}
>
> The results are presented in the following table. For validity, compared to diffusion-based and deep generative models, RW cannot simultaneously meet the conditions since it only focuses on structural information. Regarding the discrepancy with the input graphs, those generated by RW typically exhibit the largest disparity because RW lacks global awareness and is highly influenced by biases in local structure. Concerning homophily, RW yields high relative error ratios of assortativity and EI homophily index since it neglects the attributes of nodes when generating graphs.
> | Metric   | Validity | Node     | Edge   | Density | Clust. coeff. | Assortativity | EI homophily index |
> |----------|----------|----------|--------|---------|---------------|---------------|--------------------|
> | RW       | 0        | 5.170    | 8.225  | 0.698   | 0.242         | 0.351         | 1.437             |
> | SPECTRE  | 0.276    | _0.110_| 1.113  | 0.439   | 0.288         | 0.042         | 0.172             |
> | GSM      | 0.274    | 0.147    | _0.677_ | 0.457 | 0.031         | 0.043         | 0.175             |
> | EDGE     | 0.307    | 0.897    | 2.424  | 0.926   | _0.03_      |  **0.006**     | **0.025**             |
> | DiGress  | _0.375_| __0.002__| 0.725  | _0.297_| 0.259         | 0.067         | 0.275             |
> | CDGraph  | __1__    | 0.149    | __0.186__| __0.002__ | __0.022__   | _0.013_         | _0.052_             |

---

> ### Author Response · Authors · 2023-11-22
> **Responses to weakness 4~5**
>
> **W4**: Ablation study.
>
> **Response**: We apologize that the results of the ablation study could not be produced within the time limit because the interdependence among modules is complex, and removing the implementation of each module requires a bit more time. We plan to introduce three variants of CDGraph: CDGraph-coevolve, CDGraph-class, and CDGraph-plain. CDGraph-coevolve removes the co-evolution dependency module in the reverse denoising process, CDGraph-class discards the dual conditional classifier module in the reverse denoising process, while CDGraph-plain omits both modules. The ablation study will be included in the final manuscript.
>
> **W5**: The headers of some figures and tables.
>
> **Response**: As suggested, we revise the titles of tables and figures as follows:
> * Table 1 in the revised paper (previously Table 2 in the original paper): Validity and discrepancy with the input graphs of different approaches on all datasets.
> * Figure 2 in the revised paper (previously Figure 3 in the original paper): Workflow of CDGraph, where $\mathbf{E}$ denotes the edge embedding, and $\mathbf{X_{c_{1}}}$, and $\mathbf{X_{c_{2}}}$ are embeddings of nodes indicating the satisfaction of conditions $c_1$ and $c_2$.
> * Figure 3 in the revised paper (previously Figure 2 in the original paper): Co-evolution dependency incorporating social homophily and social contagion.
>   - (a) Social Homophily-based Co-evolution.
>   - (b) Social Contagion-based Co-evolution.
>
> The other figures and tables are moved to the appendix due to the space constraint.

---

> ### Author Response · Authors · 2023-11-23
> **References**
>
> **References**
>
> [1] K. Nakajima and K. Shudo, "Social Graph Restoration via Random Walk Sampling," 2022 IEEE 38th International Conference on Data Engineering (ICDE), Kuala Lumpur, Malaysia, 2022, pp. 01-14.
>
> [2] Newman, M. E. J.. "Mixing patterns in networks". Physical Review, 2003.
>
> [3] Krackhardt, David, and Robert N. Stern. "Informal Networks and Organizational Crises: An Experimental Simulation." Social Psychology Quarterly, 1988.
>
> [4] E. Lee, F. Karimi, C. Wagner, H.-H. Jo, M. Strohmaier, and M. Galesic. Homophily and minority-group size explain perception biases in social networks. Nature human behaviour, 2019.
>
> [5] N. Horsevad, D. Mateo, R. E. Kooij, A. Barrat, and R. Bouffanais. Transition from simple to complex contagion in collective decision-making. Nature Communications, 2022.
>
> [6] C.R. Shalizi and A. C. Thomas. Homophily and contagion are generically confounded in observational social network studies. Sociological Methods & Research, 2011.
>
> [7] N. Li, C. Gao, D. Jin, and Q. Liao. Disentangled modeling of social homophily and influence for social recommendation. IEEE Transactions on Knowledge and Data Engineering, 2023.
>
> [8] A. Sankar, X. Zhang, A. Krishnan, and J. Han. Inf-VAE: A variational autoencoder framework to integrate homophily and influence in diffusion prediction. In ACM WSDM, 2020.
>
> [9] H. Bisgin, N. Agarwal & X. Xu, A study of homophily on social media. World Wide Web 15, 213–232 (2012).
>
> [10] Marios Papachristou, Siddhartha Banerjee, and Jon Kleinberg. Dynamic Interventions for Networked Contagions. In ACM WWW '23.
>
> [11] Julie Jiang, Ron Dotsch, Mireia Triguero Roura, Yozen Liu, Vítor Silva, Maarten W. Bos, and Francesco Barbieri. Reciprocity, Homophily, and Social Network Effects in Pictorial Communication: A Case Study of Bitmoji Stickers. In ACM CHI '23.

---

### Official Review · Reviewer_FtFj · 2023-11-02

**Soundness:** 2 fair
**Presentation:** 3 good
**Contribution:** 2 fair
**Rating:** 5
**Confidence:** 2

**Summary:**

In response to the growing demand for social graphs in light of data scarcity and privacy concerns,  this paper introduces CDGraph, a novel conditional diffusion model for social networks. CDGraph effectively synthesizes social graphs while adhering to specified conditions.

**Strengths:**

(1) The paper is the first attempt to develop a conditional diffusion model for social networks.
(2) The process of CDGraph is introduced in detail.

**Weaknesses:**

(1)  The authors propose social homophily-based co-evolution and social contagion-based co-evolution. However, how CDGraph captures the interdependencies between these specified conditions does not seem to be clearly explained.
(2)  Figure 3 lacks relevant figure captions, making it slightly difficult to read.
(3)  The reasons for some experimental results are not explained in depth. For example, there is a lack of detailed reasons why Relative error ratios of #nodes on the BlogCatalog data set, and Clust. coeff reaches suboptimality.
(4)  The authors fail to describe its limitations and broader impacts.

**Questions:**

It seems that the comparison algorithms only consider one condition. Only the proposed CDGraph considers two conditions, and the evaluation indicator "Validity" evaluates the proportion of nodes that meet the two specified conditions? Is this evaluation indicator unreasonable enough? （in addition to the restructured DiGress）

---

> ### Author Response · Authors · 2023-11-21
> **Answers to Question 1**
>
> **Q1**: The reasonability of the validity evaluation.
>
> **Response**: We clarify that validity refers to the portion of generated graphs in which the majority of nodes meet the two specified conditions, rather than the proportion of nodes meeting these conditions. Although the baselines are not designed to control the generation process to meet the two conditions, we extend them to support graphs with nodes having two attributes as input and capable of generating graphs with nodes having two attributes. Through the evaluation of validity, we can conclude that, for baselines other than DiGress, as they do not consider meeting both conditions during generation, even if they learn from graphs with nodes having two attributes, they cannot guarantee the generation of graphs meeting both conditions. While DiGress is also extended to support two conditions, it cannot achieve high validity, indicating the need to design new methods to generate graphs that can meet both conditions.

---

> ### Author Response · Authors · 2023-11-21
> **Responses to weaknesses 1~4**
>
> **W1 and W2**: Detailed explanation of the workflow of CDGraph.
>
> **Response**:  The workflow of CDGraph is updated in Figure 2 of the revised paper (previously Figure 3 in the original paper), and we provide the detailed explanation as follows: The forward noising processes of $\mathbf{X_{c_{1}}}^{(t)}$, $\mathbf{E}^{(t)}$, and $\mathbf{X_{c_{2}}}^{(t)}$ are illustrated from above to below, and each $\mathbf{X_{c_{i}}}^{(t)} (i=1,2)$ has dependency with $\mathbf{E}^{(t)}$. The red bending arrow represents the conditional probability distribution $q(\mathbf{X_{c_{i}}}^{(t)}|\mathbf{E}^{(t)})$ of $\mathbf{X_{c_{i}}}^{(t)}$ given $\mathbf{E}^{(t)}$, and the black bending arrow represents the conditional probability distribution $q(\mathbf{E}^{(t)}|\mathbf{X_{c_{i}}}^{(t)})$ of $\mathbf{E}^{(t)}$ given $\mathbf{X_{c_{i}}}^{(t)}$.
>
> On the other hand, the reverse denoising processes of $\mathbf{X_{c_{1}}}^{(t)}$, $\mathbf{E}^{(t)}$, and $\mathbf{X_{c_{2}}}^{(t)}$ are illustrated from above to below, and the blue arrow represents the co-evolution based on social contagion; the red arrow represents the co-evolution based on social homophily.
>
> The illustrations of social homophily and social contagion are updated in Figure 3 of the revised paper (previously Figure 2 in the original paper), and we provide a detailed explanation as follows: The social homophily-based coevolution aims to denoise edge embeddings from given node embeddings of each condition (as stated in Eq. (7)); the social contagion-based co-evolution aims to denoise node embeddings of each condition from given edge embeddings (as stated in Eq. (9)).
>
> **W3**: Detailed reasons of the suboptimality of relative error ratios of #nodes and clustering coefficient on the BlogCatalog data set
>
> **Response**: Though the number of nodes in generated graphs is sampled by empirical distribution in CDGraph and GSM, which may result in slight suboptimality of CDGraph’s relative error ratio of #nodes. Nevertheless, density refers to the ratio of the number of edges to the maximum possible number of edges (which is related to the square of the number of nodes) in a graph. Hence, it is more comprehensive to simultaneously examine the relative error ratios of nodes, edges, and density. In CDGraph, the relative error ratios for both edges and density are much lower than GSM since GSM does not aim to optimize the structure discrepancy. Overall, the structure of the graphs generated by CDGraph remains very close to that of the input graphs.
>
> On the other hand, the MMD of clustering coefficients of EDGE slightly outperforms CDGraph on BlogCatalog since EDGE focuses on the guidance of degree changes, leading to smaller differences in the clustering coefficient between input graphs and generated graphs.
>
> **W4**: Limitations and broader impacts of CDGraph.
>
> **Response**: The limitations and broader impacts are presented as follows.
> * Limitations: Currently, we focus on conditional graph generation by specifying two (node-level) conditions, and we will further generalize the case of graph generation with more than two conditions, in which the conditions involve those of edge-level or pattern-level.
> * Broader impacts:  CDGraph can generate social graphs for downstream analysis, such as social influence analysis and link prediction (friend recommendation).

---

### Official Review · Reviewer_FhD6 · 2023-11-02

**Soundness:** 3 good
**Presentation:** 2 fair
**Contribution:** 2 fair
**Rating:** 5
**Confidence:** 4

**Summary:**

The proliferation of social networks has led to the demand for synthetic social graphs that mimic real ones, allowing for the analysis of specific user profiles and network structures. Existing methods, such as statistical sampling and deep generative models, have limitations in capturing dependencies across dual conditions and accounting for social homophily and social contagion phenomena in conditional social graphs. This paper introduces a novel conditional graph generative model *CDGraph*, exploiting co-evolution dependency to guide the diffusion process to capture the dependencies between the node and edge conditions, solving the problem of the limitation of social homophily and social contagion. Furthermore, the authors propose a dual conditional classifier to guide the sampling process to fulfill dual conditions while capturing the correlation between the specified conditions. The paper conducts extensive experiments on real-world networks and evaluates the performance on validity and error discrepancies of the generated graphs compared to the input graphs. Results show that the proposed model outperforms the state-of-the-art methods under different correlations among specified conditions.

**Strengths:**

- This paper astutely highlights the presence of interdependencies among correlated conditions in conditional social graphs and provides insights into the evolution of social graphs concerning social homophily and social contagion.

- This paper proposes a novel notion of co-evolution dependency to be implemented on the conditional diffusion process of *CDGraph*, which naturally integrates the structure of diffusion denoising models and the social homophily-based and social contagion-based co-evolution of nodes and edges in social graphs.

- A formally derived dual conditional classifier is proposed to guide the sampling process to fulfill dual conditions while capturing the correlation between the specified conditions, which strengthens the validity of the generated graphs.

- The evaluation in the paper focuses on assessing the validity and error discrepancies of the generated graphs in comparison to the input graphs. The terms of evaluation are thoroughly covered.

**Weaknesses:**

- I think the reason why a dual conditional social graph can be stated more clearly in the paper. The scarcity of data is not a sufficient reason to justify the need for a dual conditional social graph. Is the **dual conditional** social graph applicable to any representative downstream scenarios?

- The crucial phenomena of social homophily and social contagion of social graphs are stated in the paper. However, I am wondering if the error ratios and the MMD metrics are sufficient to evaluate the performance of the proposed model in capturing the social homophily and social contagion is not included in the paper, and I think more metrics and experiment forms may be added, as the nature of network co-evolution is the primary motivation of the proposed model.

- The network parameterization (denoiser architecture) should be provided in the paper (could be in the appendix). Is there any specified design of the network structure in light of this problem?

- The effect of the classifier guidance is not sufficiently demonstrated in the experiments. An ablation study may be needed, by comparing the performance of the proposed model with and without the classifier.

- The details of the calculations of the metrics should be provided in the appendix.

**Questions:**

Most of my concerns are raised in the Weaknesses section. Further questions are listed below:

- The dimensions of the notations in Sec 3.1 are not clearly stated, nor the exact size of input of the diffusion process. What's more, there seems to be a mistake in mixing up the $E_c$ and $E$, and the definition of $E$ is not clear. What is the meaning of "E is a one-hot encoding vector representing whether **an edge** between v_n and v_m in GC **satisfies condition c**"?

- The formulation in equation 5 does not seem to contribute to the method's design. What is the motivation for introducing the formulation?

- As the number of dimensions needs to be specified in the input of the diffusion reverse process, the number of nodes should be a human-specified parameter, which does not seem proper to be a metric for evaluating the performance.

- The proposed method does not seem to be applicable to graphs with more than 2 conditions. However, the profiles of users in a social graph often consist of multiple properties. Is there any future direction to extend the method to graphs with more than 2 conditions?

- The density metric of the CDGraph in the Twitter dataset is the worst among all the methods. Is there any explanation for this?

**Details Of Ethics Concerns:**

The proposed approach is a generative model capable of producing synthetic social graphs based on input graphs. The generated graphs are intended for downstream tasks like user profiling and network analysis. The model learns the correlation of conditions (properties) and the structure of network topologies from real-world data, which may introduce biases into the generated graph. These biases may be exacerbated in downstream tasks, resulting in unfairness.

Nonetheless, such phenomena are frequently observed in machine learning and generative models, and the authors are not obliged to address the issue in their paper.

---

> ### Author Response · Authors · 2023-11-21
> **Answers to Questions 1~5**
>
> **Q1 and Q3**: Dimensions of the input in the diffusion process.
>
> **Response**: The dimensions of the notations are stated as follows. (Suppose $N$ is the number of nodes in a graph $G$.)
> * The one-hot encoding of nodes indicating the satisfaction of condition $c_{i}$: $\mathbf{X} _{c_i} \in \mathbb{R}^{N \times 2}$.
> * The one-hot encoding of edges: $\mathbf{E} \in \mathbb{R}^{N \times N \times 2}$.
> * The transition matrix for $\mathbf{X_{c_i}}$: $\mathbf{Q}_{X _{c_i}}^{(t)} \in \mathbb{R}^{2 \times 2}$
> * The transition matrix for $\mathbf{E}$: $\mathbf{Q}_{E}^{(t)} \in \mathbb{R}^{2 \times 2}$
>
> Besides, we fix the typo and clarify that $\mathbf{E}$ is a one-hot encoding vector representing whether there is an edge between $v_n$ and $v_m$ in the definition of the conditional indication graph.
>
> However, according to previous works [1, 2], the number of nodes in the generated graph is drawn from an empirical distribution rather than specified by user input, thus becoming an indicator of whether the generated graph can effectively mimic the original graph.
>
> **Q2**: Motivation for introducing the formulation in Eq. (5).
>
> **Response**: The two equations formulated in Eq. (5) state the forward process of the nodes and edges respectively, in which the dependencies between nodes and edges are captured by the second term in both equations, and the first term in the first (second) equation models the forward processes of $\mathbf{X_{c_{i}}}^{(t)}$ given $\mathbf{X_{c_i}}^{(t-1)}$ and $\mathbf{E}^{(t-1)}$ ($\mathbf{E}^{(t)}$ given $\mathbf{E}^{(t-1)}$ and $\mathbf{X_{c_i}}^{(t-1)}$). However, due to the Markovian property of $q$, we have:
> $$q(\mathbf{X_{c_{i}}}^{(t)}|\mathbf{X_{c_{i}}}^{(t-1)}, \mathbf{E}^{(t-1)})=q(\mathbf{X_{c_{i}}}^{(t)}|\mathbf{X_{c_{i}}}^{(t-1)}); q(\mathbf{E}^{(t)}|\mathbf{X_{c_{i}}}^{(t-1)}, \mathbf{E}^{(t-1)})=q(\mathbf{E}^{(t)}|\mathbf{E}^{(t-1)})$$
>
> **Q4**: Future direction for CDGraph to support more than two conditions.
>
> **Response**: To support more than two conditions, we need to modify the probability distribution of the reverse process by augmenting the node embeddings $\mathbf{X_{c_i}}$ to indicate the satisfaction of the conditions. And the probability distribution of the classification guidance model can be modified by additionally considering the dependency between the conditions.
> We noticed that Eq. (6) can be extended by adding more feature matrices of node conditions, which can be written as follows.
> $$
> p_{\theta} ( \lbrace \mathbf{X}_{c_i}^{(t-1)} \rbrace _{i=1}^{K}, \mathbf{E}^{(t-1)} | \lbrace \mathbf{X} _{c_i}^{(t-1)} \rbrace _{i=1}^{K}, \mathbf{E}^{(t)}) = \prod _{i=1}^{K} p _{\theta} (\mathbf{X} _{c_i}^{(t-1)} | \mathbf{X} _{c_i}^{(t)}, \mathbf{E}^{(t)}) p _{\theta} (\mathbf{E}^{(t-1)} | \mathbf{E}^{(t)}, \lbrace \mathbf{X} _{c_i}^{(t-1)} \rbrace _{i=1}^{K}),
> $$
>
> where $K$ is the number of specified conditions.
> On the other hand, the dual-conditional classification guidance model can also be extended by specifying more conditions as follows:
> $$
> q (G_C^{(t-1)} | G_C^{(t)}, \lbrace c_i \rbrace _{i=1}^{K}) \propto q(c_i | G_C^{(t-1)}, \lbrace c_j \rbrace _{j \neq i}) \prod _{j \neq i} q (c_j | G_C^{(t-1)}) q (G_C^{(t-1)} | G_C^{(t)})
> $$
>
> where $K$ is the number of specified conditions, and $\lbrace c_j \rbrace _{j \neq i}$ are conditions dependent with $c_i$.
>
> **Q5**: Explanation of the performance of CDGraph’s density metric on the Twitter dataset.
>
> **Response**: We are sorry that we got the data wrong. The correct relative error ratio of density in each method is listed as follows:
> * GSM: 0.4580
> * SPECTRE: 0.4410
> * EDGE: 0.0531
> * DiGress: 0.0840
> * CDGraph: 0.2009
>
> The relative error ratio of density in EDGE and DiGress is lower than in CDGraph since EDGE focuses on guiding degree changes, and DiGress attempts to preserve graph sparsity in the diffusion process. Nevertheless, density refers to the ratio of the number of edges to the maximum possible number of edges (which is related to the square of the number of nodes) in a graph. Hence, it is more comprehensive to simultaneously examine the relative error ratios of nodes, edges, and density. In CDGraph, the relative error ratios for both nodes and edges are less than 0.05. Overall, the structure of the graphs generated by CDGraph remains very close to that of the input graphs.

---

> ### Author Response · Authors · 2023-11-21
> **Responses to weaknesses 1~2**
>
> **W1**: The justification of the need of dual-conditional social graphs (and the downstream scenarios).
>
> **Response**: The generation of dual conditional social graphs is highly valuable for social applications, as these applications often require the consideration of users' diverse social contexts and connections, which typically satisfy more than one condition in real-world applications. Moreover, they should also retain the unique characteristics of social graphs, i.e., social homophily and social contagion, to ensure that the phenomena inherent in social applications can be captured. Social homophily [6], which involves the tendency to form social connections with people who share similar profiles, plays a significant role as a primary factor in link formation [1]. On the other hand, social contagion [7,8], where behavior spontaneously spreads through a social graph, is largely influenced by network topology [2]. Since homophily and contagion are generally intertwined [3], many studies often simultaneously consider both to facilitate social applications. For instance, Li et al. [4] explore the effects of social homophily and contagion on users' behavior for social recommendation. Sankar et al. [5] leverage social homophily and contagion to accurately predict the spread of information.
>
> For downstream scenarios like social marketing, it is common to select individuals with strong influence within their social circles for word-of-mouth promotion. These social circles are parts of users' ego social graphs, preserving the essential characteristics of social homophily and social contagion to accurately capture social influence propagation. Specifically, for luxury golf club brands targeting golf enthusiasts and high-income individuals, analyzing users' influence within social circles that include friends sharing a passion for golf and possessing high incomes can help identify individuals well-suited for promoting the brands. Similarly, in link prediction, recommending friends with similar social circles is fundamental. Therefore, focusing on users' social circles characterized by a shared interest in birdwatching and frequent visits to locations in Japan ensures precise recommendations of friends involved in birdwatching within Japan.
>
> **W2**: More metrics and evaluations.
>
> **Response**: We conduct additional experiments for evaluation with assortativity [9] and EI homophily index [10], detailed as follows. Note that to demonstrate the discrepancy with the input graphs, these metrics are presented using relative error ratios.  Assume that there are two attributes associated with each node, and each node has a binary value (0/1) for each attribute. A homo-edge is an edge where the end nodes share the same value for every attribute, whereas a hetero-edge is an edge where the end nodes differ in values for at least one attribute.
> * Assortativity [9]: The ratio of the number of homo-edges $|E_{homo}|$ to the number of edges $|E|$ in the whole graph:  $$r=\frac{|E_{homo}|}{|E|}.$$
> * EI homophily index [10]: The ratio of the difference between the number of hetero-edges $|E_{hetero}|$ and the number of homo-edges $|E_{homo}|$ to to the number of edges $|E|$ in the whole graph:
> \begin{equation*}
> EI=\frac{|E_{hetero}|-|E_{homo}|}{|E|}.
> \end{equation*}
>
> The results are presented in the table below. For both assortativity and the EI homophily index, CDGraph obtains the second-lowest relative error ratios, indicating the effectiveness of the design based on social homophily and social contagion co-evolution. It's worth noting that EDGE achieves the lowest relative error ratios, as its generation process is guided by nodes' degrees, which may implicitly learn associations with node attributes from the input graphs. Despite EDGE performing well in homophily discrepancy, its validity is significantly lower than that of CDGraph, indicating that it still cannot generate appropriate dual conditional social graphs. Other baselines, lacking specific consideration for the relationship between structure and attributes, exhibit higher relative error ratios.
>
> |            | Assortativity | EI homophily index |
> |------------|---------------|--------------------|
> | RW         | 0.351         | 1.437              |
> | SPECTRE    | 0.042         | 0.172              |
> | GSM        | 0.043         | 0.175              |
> | EDGE       | **0.006**     | **0.025**          |
> | DiGress    | 0.067         | 0.275              |
> | CDGraph    | _0.013_         | _0.052_              |

---

> ### Author Response · Authors · 2023-11-21
> **Responses to weaknesses 4~5**
>
> **W3**: Denoising network parameterization.
>
> **Response**: We provide the architecture of the denoising network and elaborate the details as follows. Following DiGress [11], we construct a Graph Transformer by first extracting graph-theoretic (structural and spectral) features from node and edge embeddings and then updating the embeddings of nodes and edges, as well as graph-level features through graph transformer layers, each of which consists of a self-attention module. We extend the architecture of Graph Transformer employed in DiGress by augmenting the node embeddings since nodes in the condition indication graph are associated with two conditions (there is a 1-hot encoding for each condition). The graph-theoretic features are extracted by following the way in DiGress.
>
> **W4**: Ablation study.
>
> **Response**: We apologize that the results of the ablation study could not be produced within the time limit because the interdependence among modules is complex, and removing the implementation of each module requires a bit more time. We plan to introduce three variants of CDGraph: CDGraph-coevolve, CDGraph-class, and CDGraph-plain. CDGraph-coevolve removes the co-evolution dependency module in the reverse denoising process, CDGraph-class discards the dual conditional classifier module in the reverse denoising process, while CDGraph-plain omits both modules. The ablation study will be included in the final manuscript.
>
> **W5**: Calculations of the evaluation metrics.
>
> **Response**: The definitions of the metrics are stated as follows:
> * Validity (the higher, the better): Suppose $D= \lbrace G_1,...,G_n \rbrace$ is the set of graphs generated by a graph generation algorithm. Let $D_{val}= \lbrace G_i \in D|r_{c_1,c_2}(G_{i}) \geq 0.5 \rbrace$ denote the set of valid graphs in $D$, where $r_{c_1,c_2}(G_{i})$ is the ratio of the nodes in $G_{i}$ satisfying the specified conditions $c_1$ and $c_2$. The validity is calculated as follows:
> $$Validity = |D_{val}|/|D|$$
> * Relative error ratios (the lower, the better): Suppose $D= \lbrace G_1,...,G_n \rbrace$ is the set of original input graphs and $\hat{D}= \lbrace \hat{G}_1,...,\hat{G}_n \rbrace$ is the set of graphs generated by a graph generator based on $D$. The relative error ratio regarding a graph metric $m$ (e.g., average #nodes and #edges, average density) is calculated by: $m(\hat{G})/m(G)$.
> * MMD (maximum mean discrepancy; the lower, the better): Suppose $G$ is the original input graph and $\hat{G}$ is the generated graph. The MMD between $G$ and $\hat{G}$ is calculated by the difference between their graph metric distribution $f(G)$ and $f(\hat{G})$, where $f$ can be the degree distribution, local clustering coefficient distribution, etc.

---

> ### Author Response · Authors · 2023-11-23
> **References**
>
> [1] E. Lee, F. Karimi, C. Wagner, H.-H. Jo, M. Strohmaier, and M. Galesic. Homophily and minority-group size explain perception biases in social networks. Nature human behaviour, 2019.
>
> [2] N. Horsevad, D. Mateo, R. E. Kooij, A. Barrat, and R. Bouffanais. Transition from simple to complex contagion in collective decision-making. Nature Communications, 2022.
>
> [3] C.R. Shalizi and A. C. Thomas. Homophily and contagion are generically confounded in observational social network studies. Sociological Methods & Research, 2011.
>
> [4] N. Li, C. Gao, D. Jin, and Q. Liao. Disentangled modeling of social homophily and influence for social recommendation. IEEE Transactions on Knowledge and Data Engineering, 2023.
>
> [5] A. Sankar, X. Zhang, A. Krishnan, and J. Han. Inf-VAE: A variational autoencoder framework to integrate homophily and influence in diffusion prediction. In ACM WSDM, 2020.
>
> [6] H. Bisgin, N. Agarwal & X. Xu, A study of homophily on social media. World Wide Web 15, 213–232 (2012).
>
> [7] Marios Papachristou, Siddhartha Banerjee, and Jon Kleinberg. Dynamic Interventions for Networked Contagions. In ACM WWW '23.
>
> [8] Julie Jiang, Ron Dotsch, Mireia Triguero Roura, Yozen Liu, Vítor Silva, Maarten W. Bos, and Francesco Barbieri. Reciprocity, Homophily, and Social Network Effects in Pictorial Communication: A Case Study of Bitmoji Stickers. In ACM CHI '23.
>
> [9] Newman, M. E. J.. "Mixing patterns in networks". Physical Review, 2003.
>
> [10] Krackhardt, David, and Robert N. Stern. "Informal Networks and Organizational Crises: An Experimental Simulation." Social Psychology Quarterly, 1988.
>
> [11] C. Vignac, I. Krawczuk, A. Siraudin, B. Wang, V. Cevher, and P. Frossard, “Digress: Discrete denoising diffusion for graph generation.” ICLR, 2023.

---

### Author Response · Authors · 2023-11-21
**General response**

We sincerely appreciate the reviewers' time and insightful comments. Overall, the reviewers recognize the novelty of our work (FhD6, FtFj, QDkE, zZ91), our model design for dual conditional social graph generation (FhD6, QDkE, zZ91), and the effectiveness of our approach (FhD6, QDkE). The major concerns of the reviewers lie among the motivation justification (FhD6, QDkE, zZ91), detailed elaboration of the modeling design (FtFj, QDkE, zZ91), and additional experiments including additional metrics and baselines (FhD6, FtFj, QDkE). Specifically, we made the following changes:
* **Motivation Justification.** We enhance the motivation of incorporating dual conditions, social homophily, and social contagion in social graph generation and also provide more concrete examples to illustrate its importance.
* **Model Design.** We clarify the model design of CDGraph and update Figures 2 and 3, which illustrate social homophily and social contagion-based co-evolution and the overall framework of CDGraph, respectively, to provide a clearer understanding.
* **Additional Experiments.** We add a new baseline and two new social metrics for more comparative studies.

Accordingly, we also restate our contributions as follows.
* **Importance and Novelty.** We justify the need to generate dual conditional social graphs exhibiting fundamental characteristics such as social homophily and social contagion. To address this requirement, we develop a novel dual conditional graph diffusion model, CDGraph.
* **Sound Model Design.** We propose co-evolution dependency not only to capture the interdependence between specified conditions but also to denoise edge and node embeddings based on social homophily and social contagion, respectively. Meanwhile, we introduce a dual-condition classifier to guide the denoising process, ensuring that the discrepancy in the diffusion process and condition fulfillment can be jointly optimized.
* **Effectiveness Assessed by Extensive Experiments.** Experiments conducted on four real-world social graphs demonstrate that CDGraph outperforms graph generation approaches based on traditional, diffusion-based, and deep generative models in various social metrics.

---

### Meta-Review · Area_Chair_otfg · 2023-12-06

**Metareview:**

None of the reviewers is positive about the paper and they raise a number of concerns regarding the motivation, methodology and experimental evaluation. The authors' rebuttal did not persuade them to become positive about the ppaper. I encourage the authors to revise their paper in light of the reviewers extensive reviews and submit to another top tier venue.

**Justification For Why Not Higher Score:**

None of the reviewers is positive about the paper.

**Justification For Why Not Lower Score:**

N/A

---

### Decision · Program_Chairs · 2024-01-16

Reject